# Small-Angle X-ray Scattering Models of APOBEC3B Catalytic Domain in a Complex with a Single-Stranded DNA Inhibitor

**DOI:** 10.3390/v13020290

**Published:** 2021-02-12

**Authors:** Fareeda M. Barzak, Timothy M. Ryan, Maksim V. Kvach, Harikrishnan M. Kurup, Hideki Aihara, Reuben S. Harris, Vyacheslav V. Filichev, Elena Harjes, Geoffrey B. Jameson

**Affiliations:** 1School of Fundamental Sciences, Massey University, Private Bag 11 222, New Zealand; f.barzak@massey.ac.nz (F.M.B.); m.kvach@googlemail.com (M.V.K.); H.Kurup@massey.ac.nz (H.M.K.); 2SAXS/WAXS, Australian Synchrotron/ANSTO, 800 Blackburn Road, Clayton, VIC 3168, Australia; timoryan@ansto.gov.au; 3Department of Biochemistry, Molecular Biology, and Biophysics, University of Minnesota, Minneapolis, MN 55455, USA; aihar001@umn.edu (H.A.); rsh@umn.edu (R.S.H.); 4Howard Hughes Medical Institute, University of Minnesota, Minneapolis, MN 55455, USA; 5Maurice Wilkins Centre for Molecular Biodiscovery, Auckland 1142, New Zealand

**Keywords:** APOBEC, SAXS, APOBEC3, APOBEC3B, dimer, virus restriction, drug resistance, APOBEC inhibitors, cancer evolution

## Abstract

In normal cells APOBEC3 (A3A-A3H) enzymes as part of the innate immune system deaminate cytosine to uracil on single-stranded DNA (ssDNA) to scramble DNA in order to give protection against a range of exogenous retroviruses, DNA-based parasites, and endogenous retroelements. However, some viruses and cancer cells use these enzymes, especially A3A and A3B, to escape the adaptive immune response and thereby lead to the evolution of drug resistance. We have synthesized first-in-class inhibitors featuring modified ssDNA. We present models based on small-angle X-ray scattering (SAXS) data that (1) confirm that the mode of binding of inhibitor to an active A3B C-terminal domain construct in the solution state is the same as the mode of binding substrate to inactive mutants of A3A and A3B revealed in X-ray crystal structures and (2) give insight into the disulfide-linked inactive dimer formed under the oxidizing conditions of purification.

## 1. Introduction

The human APOBEC3 (A3) family of cytidine deaminases acts in various roles within the innate immune system, largely by deaminating single-stranded DNA (ssDNA, Scheme 1A) [1,2]. In humans there are seven A3 enzymes, APOBEC3A-APOBEC3H (A3A-A3H, excluding A3E), which have diverged and expanded from a single enzyme to provide protection against a range of exogenous retroviruses, DNA-based parasites, and endogenous retroelements [1,3,4]. However, misregulation of several A3 family members, in particular the mutagenic actions of A3A, A3B, and A3G, are exploited by viruses (including SARS-CoV-2) [5,6,7] and cancer cells to enhance their rate of evolution, leading to detrimental outcomes by their escaping adaptive immune responses and becoming drug resistant [8,9,10,11,12,13,14].

The A3 members exist as either single-domain enzymes (A3A, A3C, and A3H) or double-domain enzymes (A3B, A3G, A3D, and A3F) made up of two tandem homologous domains with a short flexible linker [15,16,17]. In double-domain A3 enzymes, the carboxy-terminal domain (CTD) encompasses the catalytic activity, whereas the amino-terminal domain (NTD) has little to no catalytic activity, but is reported to be important in binding ssDNA substrates, enhancing enzyme activity above that of the CTD domain alone [18,19,20,21,22,23,24]. The A3 enzyme domains show a similar conserved globular fold, consisting of a hydrophobic core formed by five β-strands flanked by six α-helices that are connected through flexible loops (Figure 1A) [25,26,27]. The active-site (Figure 1B) is highly conserved among the A3 (and single-nucleo(s/t)ide-accepting cytosine deaminases (CDAs)) [16,28,29,30], comprising a single zinc (Zn^2+^) ion and a hallmark zinc-dependent deaminase domain (ZDD) motif (His-x-Glu-x_23–28_-Pro-Cys-x_2_-Cys, where x is any amino acid residue), where the histidine and two cysteines (sometimes four cysteines in CDAs) of this motif coordinate the Zn^2+^ ion into the active-site; the catalytic glutamic acid, Glu72 in A3A and Glu255 in A3B, is an essential general acid/base to mediate the reaction [16,28,29,30].

Differences between the A3 catalytic domain structures are predominantly found in loops 1, 3, and 7, which surround the active site, reflecting differences in the intrinsic ssDNA sequence preference, physiological roles, and potential to oligomerize [31,32,33,34]. Several studies have suggested that oligomerization of some A3 proteins (A3B, A3D, A3F-H) [35,36,37,38,39,40,41,42,43,44,45,46,47] may be involved in regulating catalytic activity by forming inactive conformations [25,48,49,50,51,52]. Therefore, structural studies on how these enzymes associate will help in understanding the role, if any, of A3 oligomerization. Recently, crystal structures were reported for ssDNA-bound to catalytically inactive A3A-E72A (PDB: 5SWW, 5KEG), the E255A-inactivated A3B_CTD_-QM-∆L3-AL1swap chimera (hereafter termed A3B_CTD_*; PDB: 5TD5) and A3G_CTD_ (PDB: 6BUX) [30,53,54]. Inactivation of the enzyme was necessary to prevent substrate deamination in those systems. The ssDNA has a pronounced kink at the target dC and preceding nucleobase, adopting a U-shaped conformation on binding, seen also in solution [55].

In the case of A3A-E72A (5SWW), association of a neighboring molecule with the 6-mer ssDNA is observed for two of the four crystallographically independent molecules in the crystal structure. For A3A-E72A (5KEG), which binds a 5-mer ssDNA, Cys64 on extended loop 3 forms an intermolecular disulfide link with its symmetry-related mate and in the ssDNA-free structure (4XXO) a pair symmetry-related His56 is bridged by Zn^2+^ to form a dimer. Zn^2+^ also features in linking pairs of molecules of a A3G_CTD_ construct where Cys308 is bound to a quinone derivative (3V4J) [56] with Cys261 from one molecule linked via Zn^2+^-OH to His248 and His250 of a neighboring molecule. In the case of 9-mer ssDNA bound to inactive A3G_CTD_-E259A, both 5′- and 3′-ends form non-Watson-Crick base pairs with neighboring A3G_CTD_-ssDNA complexes to form a linear polymer in the crystal. The question is which, if any, of these intermolecular associations observed in the crystalline state are relevant in the solution state, let alone which of these associations, if any, might be physiologically relevant. Small-angle scattering, especially small-angle X-ray scattering (SAXS), can shed light on solution state associations. 

So far there have not been any experimentally based models or structures of catalytically active A3 bound to ssDNA. Several studies have reported that ssDNA substrates bind weakly to inactivated A3 catalytic domains (medium-high µM range) [57,58,59,60] and reintroduction of the catalytic glutamate increased the strength of binding [59]. Therefore, to enhance the binding affinity and prevent deamination during experiments, we have studied the binding to active A3 enzymes of our recently characterized ssDNA-based A3 inhibitor containing 2′-deoxyzebularine (dZ, Scheme 1B). The sequence 5′-ATTT-dZ-ATTT (abbreviated as dZ-oligo) has a low (8–11) µM binding affinity for active A3 enzymes, including A3B_CTD_* and A3B_CTD_-DM [59,61]. The active A3B_CTD_*/dZ-oligo complex was analyzed using small-angle X-ray scattering (SAXS) to elucidate the changes in SAXS profiles upon ssDNA binding and subsequently to derive solution-state SAXS structural models to complement the solid-state X-ray structures. To facilitate comparison with X-ray single-crystal structural data and link with activity data, the A3B_CTD_* construct, rather than wild-type A3B_CTD_, was used in our SAXS study. This work provides a platform for further solution-state structural studies of full-length active A3 enzymes bound to ssDNA-based inhibitors.

## 2. Materials and Methods

### 2.1. Synthesis of 2′-Deoxyzebularine (dZ) Containing Oligodeoxynucleotide

The 2′-deoxyzebularine (dZ) containing oligodeoxynucleotide (5′-ATTT-dZ-ATTT, dZ-oligo) was prepared as described previously [59]. dZ oligo shows a low micromolar potency (*K*_i_ ~ 7.5 ± 1.7 µM) for A3B_CTD_-QM-ΔL3-AL1swap (abbreviated A3B_CTD_*) [59].

### 2.2. Protein Expression and Purification of A3BCTD* Protein

The catalytically active A3B_CTD_* was expressed and purified as described previously [59,61,62]. This variant contained four substitution mutations (F200S, W228S, L230K, F308K), the truncation of loop 3, and the transplant of loop 1 from A3A. Purification of the A3B_CTD_* variant resulted in two major peaks eluting off the size-exclusion chromatography (SEC) column in a 50 mM Tris-HCl pH 7.4, 0.5 M NaCl, and 2 mM β-mercaptoethanol (β-ME) buffer, which were collected and examined in this study.

### 2.3. Fluorescence-Based Deaminase in-Gel Activity Assay

A previously reported in vitro DNA deamination in-gel based assay [56,62] was used to assess the deamination activity of purified A3 proteins. Briefly, the fluorescently tagged oligonucleotide (TC-3′ 6-FAM-oligo, 5′-ATTATTATTATTCAAATGGAT TTATTTATTTATTTATTTATTT-fluorescein) and the purified proteins were buffer matched by diluting to the required concentrations using a 1× HEPES pH 7.4 buffer (10 mM HEPES-KOH, 50 mM NaCl at pH 7.4). In a PCR strip, reactions were setup using 5 µL of a 1.6 µM TC-3′ 6-FAM oligo mixed with 5 µL of 10 µM of the purified protein (final concentrations of 800 nM DNA to 5 µM protein). The reaction mixtures were then subjected to temperature cycling in a thermocycler (Eppendorf), first incubating for an hour at 37 °C, followed by incubation at 95 °C for 3 min, then cooling to 12 °C. The reaction mixtures were then treated with 1 µL of 120 nM uracil-DNA glycosylase (UDG), to cleave the uracil bases in the oligonucleotide) (New England Biolabs, Ipswich, MA USA) and incubated for a further 10 min at 37 °C. Following UDG treatment, 1.2 µL of a 1 M NaOH was added to the mixture and incubated for 5 min at 98 °C to stop the reaction and cleave DNA with an abasic site. To separate and resolve the ssDNA reaction products, denaturing Tris/Borate/EDTA (TBE)-Urea polyacrylamide gels for gel electrophoresis (PAGE), prepared by standard methods [63], were used.

### 2.4. Small Angle X-ray Scattering (SAXS) Setup

Small-angle X-ray scattering (SAXS) was used to provide structural information and low-resolution models of the active A3B_CTD_* protein and its complex with the dZ-oligo in solution. The use of a co-flow system coupled to a size-exclusion chromatography-SAXS system (SEC-coflow-SAXS) was used to resolve sample species, buffer match, and prevent sample degradation due to radiation damage, while additionally monitoring the elution profile by UV detection [64]. In co-flow, the sample flows into the center of the capillary cell while a matched buffer encases the sample preventing protein denaturation due to intense X-radiation depositing on the cell walls, which also enhances the signal to noise ratio [65].

Measurements were conducted at the Australian Synchrotron on the SAXS/WAXS beamline equipped with a Pilatus-2 1M detector. SAXS measurements were obtained at 25 °C, using a camera length of 1.6 m and frames were taken at one second intervals. Samples of A3B_CTD_* (0.5–10 mg/mL) in buffer (50 mM citrate-phosphate, pH 5.5, 200 mM NaCl, 2 mM β-ME, 200 μM sodium trimethyl-silyl-propane-sulfonate (DSS), 10% glycerol) with and without ssDNA at varying ratios were loaded in a 96-well plate (50 μL) and degassed for 30 min under mild vacuum in a degassing station chamber (TA Instruments). Scattering data were obtained using co-flow SEC-SAXS mode, samples were injected onto a pre-equilibrated SEC column (SEC column Superdex 75 Increase 3.2/300 GL) then run at a flow rate of 0.2 mL/min while near-simultaneously collecting SAXS data of the sample per frame. The sample’s elution profile was monitored by UV-visible spectroscopy at various wavelengths (280, 260, 220, and 480 nm), There was a slight lag of 1.05 s/frame between monitoring the elution profile by UV-visible spectroscopy and the recording of SAXS of the sample; this lag was accounted for during analysis.

### 2.5. Analysis of SAXS Data

The raw two-dimensional scattering data were processed through a number of steps. First, the raw SAXS data frames were reduced by intensity normalization (using scattering by water to place data on an absolute scale), background subtraction, and scattering vector calibration using SCATTERBRAIN 2.82 program (http://archive.synchrotron.org.au/aussyncbeamlines/saxswaxs/software-saxswaxs). A SAXS profile plot was then derived by plotting the normalized integrated intensity of the scattering signal against the frame number using CHROMIX (ATSAS 2.8.3 suite [66]) (see Scheme 2). If the profile indicated that the sample eluted as a homogeneous single species off the SEC column, then the scattering data were processed into a 1D-scattering curve. However, if the sample contained partially unresolved multiple species, then its scattering data was first deconvoluted before proceeding with further analysis. Simple deconvolution involved using SVD/EFA BioXTAS RAW [67], where the singular value decomposition (SVD) function defined the number of components in the sample (referred to as eigenvalues). Then the evolving factor analysis (EFA) method was utilized to define the boundaries and extract the scattering curves of each component.

The frames of interest were first averaged and subtracted from the buffer frames using CHROMIX. Then the processed scattering data were transformed into a 1D-scattering curve using the PRIMUSQT program from ATSAS 2.8.3 suite [66], where log of scattering intensity (log(*I*)) was plotted against the scattering vector *q* = 4π*λ*^−1^sin*θ*, in which 2*θ* represents the scattering angle, and *λ* defines the x-ray wavelength of 1.0332 Å) (Scheme 2).

Characteristic parameters can be retrieved from the scattering pattern of the samples that describe the homogeneity, fold, size, and overall shape of the sample (Scheme 3). Analysis was conducted using programs within the PRIMUSQT ATSAS 2.8.3 suite [66]. Initially, the 1D-scattering curve is converted into a double logarithmic plot to highlight that the low-*q* data has an artefact free profile, a plateau, consistent with a monodisperse protein sample. Plotting low *q* data using the Guinier distribution analysis (log(*I*) vs *q*^2^) through the AUTORG method allows estimation of the radius of gyration (*R*g) and the extrapolated intensity at zero scattering angle *I*(0), describing the overall size of the molecule. The assumption of a globular shape for the Guinier plot is valid when *qR*g ≤ 1.3 (denoted *q·R*g max) and the Guinier analysis is linear, which is also consistent with a profile of a monodisperse protein sample. Next, the data are converted into a Kratky plot (*I·q*^2^ vs *q*) to assess the shape and fold of the molecule. In addition, the Kratky plot provides information regarding the oligomeric state of the molecule [68]. An indirect inverse Fourier transformation of the scattering data performed using AUTOGNOM results in the pairwise distribution function *P*(*r*) curve, which represents the distribution of interatomic distances (*r*) within the molecule. The molecule’s maximum diameter (*D*_max_) can be determined from the *P*(*r*) curve as *P*(*r*) approaches zero at *r* >> 0 (Scheme 3).

Furthermore, the *R*g and *I*(0) can be accurately calculated from the *P*(*r*) curve using all the experimental data, unlike the Guinier analysis, which uses only a small subset at low *q*. The excluded particle volume (also termed Porod volume, *V*) is calculated through the DATPOROD program using *I*(0) values attained from the *P*(*r*) plot (Scheme 3). The Porod volume (*V*) can then be used to directly estimate the molecular weight (MW in Daltons) of the solute (MW ≈ *V* (in Å^3^)·(average protein density ~1.1 g·cm^−3^)·*N*_A_(in mol^−1^)·1 × 10^−24^ Å^3^/cm^3^) ≈ *V*·0.6) [69] providing valuable information about the oligomeric state of the molecule (Scheme 3). Furthermore, the MW can also be calculated from *I*(0) if the concentration is accurately known using a previously described method [70].

Low-resolution 3D-models can then be computed through ab initio shape restoration using the DAMMIF program by applying restraints of biophysical parameters attained from the 1D scattering curve. The assumption that scattering by the oligonucleotide can with negligible error be treated as scattering by the protein is justified as follows. The 9-mer oligonucleotide (formula; C_89_H_125_N_24_O_63_P_9_) used has intrinsic scattering of X-rays, *F*_000_, of 1450 e^-^; the protein (formula C_1102_H_1632_N_302_O_305_S_13_Zn) has *F*_000_ of 13,036 e^-^ (give or take a few electrons of water tightly associated with the protein, and to a lesser extent with the partly buried oligonucleotide). Now, assuming equal volumes per non-hydrogen atom and reconfiguring the oligonucleotide to have the protein composition of the A3B_CTD_, gives then a difference in scattering of just 256 e^-^ more for the oligonucleotide than if it had been protein. This difference is insignificant in comparison to the total scattering of 14,486 e-. However, it is important to note that the total scattering of the oligonucleotide is highly significant at just over 1/10th that of the protein and thus is potentially observable. Thus, for calculation of molecular envelopes protein-only was assumed. The DAMMIF program first generates several models, which are then averaged (DAMAVER) and are further filtered by cut-off volume constraints based on derived SAXS analysis parameters (DAMFILT) to produce the dummy filled models. The normalized spatial discrepancy (NSD) score quantitatively measures the similarities between the generated set of 3D envelope models, where NSD ≤ 0.9 is an acceptable variance [71,72]. These envelope models can be superimposed with atomic models to assemble a high-resolution model. Lastly, rigid body modelling is conducted using FoXS [73,74] and CRYSOL [66] (corrected for standard error) by comparing the experimental scattering data to the back-calculated 1D-scattering profiles of atomic structures or model structures, to validate the model. From these programs a fitting parameter termed the Chi^2^ value (also called χ^2^) can be obtained. Chi^2^ gives a measure of the discrepancies between the experimental scattering data and the back-calculated 1D-scattering profiles of atomic structures; a Chi^2^ equal to one would indicate a perfect fit.

## 3. Results

### 3.1. Methodology of Investigation: Using SAXS to Elucidate the Solution-State Structure of A3B_CTD_* in Complex with a ssDNA Inhibitor

Until recently, the structure of the A3B_CTD_-ssDNA complex was unknown. Structural studies conducted using an inactivated A3B_CTD_ construct (A3B_CTD_*-E255A: A3B_CTD_-QM-ΔL3-AL1swap-E255A chimera), where E255A is an inactivating mutation of the ZDD motif, along with a ssDNA substrate yielded the first visualization of an inactivated A3B_CTD_-ssDNA complex (PDB: 5TD5) [30]. Unlike previous studies, our current study examines an active A3B_CTD_* bound to our dZ-containing oligodeoxynucleotide (dZ-oligo) [59] in which the target dC in the preferred A3B recognition motif was substituted with dZ, a known inhibitor of cytidine deaminase (CDA) [75,76,77]. The presence of glutamic acid in the active site of the protein is essential for activation of dZ through protonation of N3 atom of the nucleobase and concurrent nucleophilic addition of the zinc-bound OH^-^/H_2_O to the C4 atom of dZ, which converts dZ into a tetrahedral transition-state analogue of cytidine deamination. These dZ-oligos allow us to study complexes formed by active A3 enzymes with ssDNA. Using active protein, the previously described A3B_CTD_*, and the 9-mer dZ-oligo, we have obtained from SAXS data the first model of an active enzyme with ssDNA, allowing comparisons to the previously published crystal structure of inactivated A3B_CTD_*-E255A with a 5-mer ssDNA substrate (PDB: 5TD5) [30].

To ensure a homogeneous sample and to reduce background noise, measurements used size-exclusion chromatography (SEC) co-flow-SAXS to examine the samples in solution. A3B_CTD_* eluted from the column as a single peak producing a good scattering curve (Figure 2A and Appendix A in Appendix A). Therefore, this variant was selected to study its conformation in solution with and without dZ-oligo.

### 3.2. SAXS Profile Analysis of A3B_CTD_* Alone and in Complex with dZ-Oligo in Solution

Initial assessment of the SEC elution profiles showed that the A3B_CTD_* protein largely eluted as a single peak with a peak maximum at around 648 s (Figure 2A), whereas the dZ-oligo eluted at around 662 s (Figure 2B). Overlaying the *R*g traces onto the elution profiles revealed that the protein had an average Rg of ~18.7 Å, whereas the dZ-oligo had an *R*g of ~9.6 Å. Moreover, both samples were monodisperse as indicated by the constant plateau of *R*g values across the elution peak, as seen in Figure 2A,B.

On adding the dZ-oligo to A3B_CTD_* (Figure 2A) at a ratio of one-to-one (protein to ssDNA) (highlighted with blue bar, Figure 2C), an increase of ~0.3 absorbance units (AU) in the protein peak maximum was observed (blue arrow, Figure 2C). This increase directly correlated with a decrease of ~0.3 AU of the dZ-oligo peak maxima (green arrow in Figure 2B,C). This change in absorbance implied that the dZ-oligo was bound to A3B_CTD_*. Consistent with these observations, the *R*g of the protein increases from a value of ~18.7 Å (Figure 2A) to ~19.5 Å in the presence of the oligo (Figure 2C–E), which also causes the protein to elute from the SEC column earlier (after ~620 s, Figure 2C–E). At a ratio of one-to-two or one-to-four (protein to oligo), the protein peak maximum increased by ~0.4 Au (blue arrow, Figure 2D,E). This indicated that the A3B_CTD_* protein was saturated with the oligo at ratios of around one-to-two (protein to oligo) or higher. Therefore, a one-to-two ratio of protein to oligo was used to ensure near-complete occupation of the A3B_CTD_* binding sites by the dZ-oligo.

The raw scattering data of A3B_CTD_*, dZ-oligo, and A3B_CTD_*/dZ-oligo samples were processed and plotted as SAXS profile plots along with the *R*g trace (see Figure 3A). The dZ-oligo resulted in an *R*g ~ *9*.5 Å (Figure 3A), consistent with having a sequence with nine nucleotides. However, due to its small size and limited resolution of SAXS (*R*g > 10 Å), further analysis of dZ-oligo was not made. On the other hand, the scattering corresponding to the A3B_CTD_* protein had a steady value of *R*g across the SAXS profile plots for both A3B_CTD_* (frames 610–660, Figure 3A) and A3B_CTD_*/dZ-oligo (frames 570–635, Figure 3A), indicating that both species were homogeneous. Therefore, these frames were averaged and transformed into 1D-scattering curves as described in Appendix A. From the 1D-scattering curves, model-free biophysical structural parameters were extracted that describe the molecules’ configuration in solution (listed in Table 1 and displayed in Figure 3).

Assessment of SAXS data for A3B_CTD_* and the A3B_CTD_*/dZ-oligo complex using the Guinier distribution showed that both samples had a good fit to the linear regression of log *I*(*q*) versus *q*^2^ at low scattering angles, indicating that the sample profiles were not aggregated (Figure 3F). Moreover, at low *q* values the double logarithmic plot, log *I*(*q*) versus log *q*, plateaued, consistent with monodispersed samples (Figure 3C). Estimation of *R*g and *I*(0) values from the Guinier plot (Figure 3F) agreed well with those derived over all data from the independent pair distribution function *P*(*r*) versus *r* for both samples (listed in Table 1), further confirming the quality and relative size of the solutes. The Kratky plot, *q*^2^·*I*(*q*) versus *q*, exhibited a bell-shaped peak at low *q* (peak maxima at *q* ~ 0.1 Å^−1^) that then converged to the scattering axis at the higher *q* range (Figure 3D), indicating that the molecules were well-folded and globular, with not much flexibility. As expected, the *R*g, *D*_max_, and Porod volume increased slightly upon formation of the A3B_CTD_*/dZ-oligo complex in comparison to the ligand-free A3B_CTD_* (as displayed in Table 1). Furthermore, the *P*(*r*) plot displayed a symmetrical curve (Figure 3E), which implied that the molecule (both with and without dZ-oligo) forms a compact near-spherical shape, consistent with results observed using the Kratky plot. These parameters were then used to calculate the molecular weight (MW). From the Porod volume and *I*(0) MWs of ~23 and 21 kDa, respectively, were calculated for the A3B_CTD_*. These MW estimates are comparable to the known MW of A3B_CTD_* of ~22 kDa [30,62]. The A3B_CTD_*/dZ-oligo sample had slightly larger MWs and increase in the overall subunit ratio (summarized in Table 1), which further supports that a protein-ssDNA complex was formed.

### 3.3. SAXS Model Structures of A3B_CTD_* Alone and in Complex with dZ-oligo

Envelope models of the A3B_CTD_* and the A3B_CTD_*/dZ-oligo complex were generated from the experimental scattering curves (Figure 4A,B) using ab initio shape restoration. Parameter information derived from the scattering curves of ten ab initio models was generated for each sample assuming *P*1 symmetry, as these samples were determined to be monomeric. The envelope models were filtered into a single model using DAMAVER and DAMFILT and used for atomic modelling, as described in Appendix A. A mean NSD score of 0.559 for the A3B_CTD_* envelope and of 0.826 for the A3B_CTD_*/dZ-oligo complex envelope indicated a very good to acceptable consistency [71,72], respectively, for the generated ensembles as summarized in Table 2.

The published X-ray crystal structures 5CQI [62] (inactive A3B_CTD_-QM) and 5TD5 [30]) (inactive A3B_CTD_*-E255A/5′-TTCAT complex) were manually docked into the experimentally derived SAXS envelope models (Figure 4). To match the size of our 9-mer oligo (dZ-oligo), four nucleotides were added using PyMol [78] onto the existing 5-mer oligo observed in the 5TD5 structure to give 5′-ATTTCATTT. The resulting structure is termed 5TD5* below. The A3B_CTD_* envelope model exhibited a globular shape with a slight concave groove (Figure 4C). On the other hand, the A3B_CTD_*/dZ-oligo envelope had an additional unoccupied electron density that resembled an arm near the groove, which was less prominent (Figure 4D). This empty electron density region was presumed to be part of the dZ-oligo indicating the location of the active site. Superimposing these low-resolution dummy models with the atomic structures (5CQI or 5TD5* [30,62]) resulted in a good visual fit, indicating that the derived SAXS envelope models corresponded well to the published atomic structures [30,62]. Based on the superimposed DNA bound A3B_CTD_ atomic structure (5TD5* [30]) and the unoccupied electron density observed in the envelope model of the A3B_CTD_*/dZ-oligo sample, it can be concluded that the dZ-oligo bound to the A3B_CTD_* protein in a similar manner as reported for the substrate, where the ssDNA binds in a U-shaped conformation [30].

Interestingly, the 3′-end of the dZ-oligo bound to A3B_CTD_* appeared to be loosely attached and conformationally flexible, as indicated by the shape of its protrusion from the protein (Figure 4D), consistent with reports of A3-ssDNA structures [30,55]. Nevertheless, to validate the generated SAXS models, rigid body modelling was performed by comparing the back-calculated 1D scattering profiles of the atomic structure models (PDB; 5CQI and 5TD5*) to the experimentally derived scattering data using FoXS [73,74] and CRYSOL [66]. The scattering data from both samples showed a good fit to the corresponding X-ray models, as summarized in Table 2 and illustrated in Figure 4A and B. In contrast, when the ligand-free A3B_CTD_* SAXS data was cross-compared to the back-calculated profile of the ssDNA-bound A3B_CTD_* atomic structure (5TD5*) or the A3B_CTD_*/dZ-oligo SAXS data was cross-compared to the ligand-free A3B_CTD_-QM-E255A atomic structure (5CQI), poor fits were observed (see Appendix A in the Appendix A). This confirms that the X-ray crystal structures (PDB 5CQI and 5TD5 with extended ssDNA) describe well the structures of the ligand-free A3B_CTD_* and the A3B_CTD_*/dZ-oligo complex in solution, respectively. We can also conclude that the shape of the A3B_CTD_*/ssDNA complex did not change significantly when the inactivating mutation (E255A) was reversed.

### 3.4. Multimerization of A3B_CTD_* in Solution

The multimerization of A3 proteins (A3A, A3B, A3D, A3F-H) has been reported to be mediated through surface interactions or through RNA interactions [47,51,52,79]. Several studies have indicated that higher oligomeric states may regulate the A3s’ catalytic activity by forming inactive conformations [25,48,49,50,51,52]. However, the biological significance of multimerization remains unclear, especially as higher concentrations of A3 favor oligomerization but contrarily, if this is the result of expression due to pathogenic attack, such oligomerization for which evidence for enhanced activity is scant, is scarcely conducive to countering the attack. During purification of the catalytically active A3B_CTD_* protein it was noted that two predominant peaks eluted off the SEC column, a peak eluting following ~90 mL of buffer, which corresponded to the monomeric A3B_CTD_* species (discussed earlier in Figure 2) and another larger species eluting off the SEC column earlier, following ~70 mL of buffer (peak 1, Figure 5A). Fractions from both peaks visualized on a reducing SDS-PAGE gel revealed that the eluants had the same molecular weight of ~22 kDa (Figure 5B) and both species corresponded to an A3B_CTD_ protein (verified using mass spectrometry analysis).

In addition, both species were catalytically active (confirmed qualitatively by an in vitro DNA deamination in-gel assay [62], see Appendix A in the Appendix A). Based on the position of the SEC elution profile, we conclude that peak 1 corresponds to homo-dimeric A3B_CTD_* (with a molecular weight of ~44 kDa, referred to as ‘A3B_CTD_* dimer’). This contrasts with the other A3B_CTD_ variants we reported in previous studies [59,61,80], A3B_CTD_-QM-ΔL3, A3B_CTD_-QM-ΔL3-E255A, and A3B_CTD_-DM, which eluted only as monomeric species. Previous reports have suggested that A3A exists as both a monomer and a dimer in vitro [52,53]. The observed multimerization of the A3B_CTD_* variant may be facilitated by the incorporation of loop 1 from A3A into the A3B_CTD_, consistent with the A3B_CTD_* adopting some characteristics of A3A [81]. Examination of the solution structure of the A3B_CTD_* dimer using SAXS may shed some light on its assembly and potential function.

The dissociation behaviour of the A3B_CTD_* dimer was examined to determine if there was an equilibrium between the two multimeric states (dimeric and monomeric forms) of A3B_CTD_*. First, the A3B_CTD_* dimer was concentrated to approximately 5 mg/mL, then re-examined using SEC-FPLC. To our surprise, the concentrated A3B_CTD_* dimer sample eluted off the SEC column as two peaks. The majority of the sample ~ 93% eluted off the column in the A3B_CTD_* dimeric form, whereas a minor peak (~7%,) corresponding to the monomeric A3B_CTD_* species was noted (peak 2, Figure 6A). In parallel, when the monomeric A3B_CTD_* sample (10 mg/mL) was concentrated we observed that the majority of the protein eluted as a monomer (~96%), while a small percentage ~ 4% eluted as the larger dimeric species (see Appendix A in Appendix A, small peak at 580 s). These findings suggested that a true equilibrium between the dimeric and monomeric A3B_CTD_* species was unlikely, and that both dimer and monomer species are mostly stable. To determine if the degree of dissociation of the dimer changes as a function of concentration, the A3B_CTD_* dimer was additionally studied under dilute concentrations (0.5, 1.25, and 2.5 mg/mL). Inspection of the elution profiles illustrated that for all the concentrations tested, the A3B_CTD_* dimer resulted in the same relative amount of monomeric A3B_CTD_* (7%–8%, observed at peak maximum around 710 s, Figure 6B), comparable with previous observations shown in Figure 6A.

The presence of surface-exposed cysteine residues, Cys239 and Cys354, in A3B_CTD_* (PDB: 5CQI) raised the possibility that a disulfide-bridged homo-dimeric species may have formed under oxidizing conditions (where the reducing SDS-PAGE showed that the dimer travelled in a similar manner as the monomeric A3B_CTD_* species). Initially, purification of the A3B_CTD_* dimer was attempted through SEC-FPLC in reducing conditions. However, this larger species remained intact, signifying that under non-denaturing conditions of SEC-FPLC (Figure 6A) (as contrasted with SDS-PAGE, Figure 5B) this conformation remained stable and the disulfide bridge (if involved in dimerization) must be buried and effectively inaccessible to the reducing agent. As the disassembly of the A3B_CTD_* dimer was not triggered by dilution, and this dimer’s stability was maintained in solution, self-association of the A3B_CTD_* monomeric subunits appears to be mediated by a disulfide linkage.

### 3.5. Interface Prediction of the A3B_CTD_* Dimer

Assessment of the potential binding interface region between two A3B_CTD_* subunits was investigated using PRISM 2.0 webserver [82,83], which uses a prediction-based algorithm with known interacting protein complexes as the template interface dataset and models the interface complex of the target proteins. The favorability of a given binding reaction is reported by the Gibbs free energy (Δ*G*), where the more negative Δ*G*, the more favourable is the formation of the dimer. Under the notion that loop 1 from A3A may influence dimerization of A3B_CTD_*, the dimerization potential of the A3B_CTD_* and A3B_CTD_-QM-ΔL3 variants were compared using PRISM 2.0 webserver [82,83] as they only differed by the swap of loop 1. The generated A3B_CTD_* interface models were found to have much lower Δ*G* values (Δ*G* ~ −38 and −18 kcal/mol, see Appendix A in Appendix A) in comparison to the interface models of the A3B_CTD_-QM-ΔL3 variant (Δ*G* ~ −13 and −3 kcal/mol, Appendix A in Appendix A), indicating that the A3B_CTD_* construct had a much greater dimerization potential than the A3B_CTD_-QM-ΔL3 construct. As A3B_CTD_* differed from the A3B_CTD_-QM-ΔL3 construct only by replacement of its loop 1 by the corresponding loop from A3A, the presence of this loop has directly influenced dimerization, consistent with experimental observations, where only the A3B_CTD_* variant resulted in the elution of a larger dimeric species (Figure 5).

Two potential dimerization models of A3B_CTD_* (termed model 1 and model 2) were predicted (Figure 7). The interface of each model was predicted to be largely formed through electrostatic interactions that accordingly stabilized the structural arrangement of a dimeric molecule (refer to Appendix A, respectively, in Appendix A). The interface of model 2 was noticeably less favourable (Δ*G* ~ −18 kcal/mol) than that of model 1 (Δ*G* ~ −38 kcal/mol) and was predicted to associate primarily through loops 3 and 5, which form part of the active-site cavity. This model, which occludes the binding site, appears inconsistent with the apparent observed activity of A3B_CTD_* dimer. As the surface of the active site is highly positively charged due to a histidine and a stretch of arginines, the interactions between residues forming this interface maybe weaker and less stable due to electrostatic repulsion (Figure 7B,D). On the other hand, the interface of model 1 occurred through interactions slightly away from the active site (Figure 7A,C).

Self-association between residues in β1, α1, α2, and loop 1 of the two molecules places in close proximity a pair of cysteines (Cys239 on beta sheet 2, β2) that would further stabilize the interface by forming a disulfide bond (see Figure 7A,C). Assessment of these models using an alternate webserver PISA [84,85], which appears to account better for compensating interactions of protein side chains with water, indicated that both of these interfaces form through weak interactions such as hydrogen bonds and salt bridges (Δ*G* < 0) (see Appendix A, respectively for model 1 and model 2 in Appendix A). Thus, the predicted disulfide bond of model 1 appears essential for structural stability of the A3B_CTD_* dimer. Further verification of this model through SAXS measurements was therefore sought.

### 3.6. SAXS Model Structure of the A3B_CTD_* Dimer

The A3B_CTD_* dimer was examined using SAXS (Figure 8) to confirm the hydrodynamic size and gain insight into the relative structure of the protein. The *R*g trace was overlaid onto the SAXS profile plot, indicating that homogeneous species were largely present across the sample peak with a *R*g of ~24 Å (frames 540–615, Figure 8A). The larger *R*g reinforced that this A3B_CTD_* species was in fact bigger than the monomeric A3B_CTD_* species (*R*g ~ 18 Å, Figure 3). The Guinier plot showed signs of slight sample aggregation at very low scattering angles (*q* < 0.0165 Å^−1^). Therefore, the first few points were removed to derive a linear dependency of the low *q* data in the Guinier plot (Figure 8C). The Kratky plot (Figure 8D) showed that the sample was a well-folded globular protein. This plot displayed a bell-shaped curve at low *q*, but at higher scattering angles (*q* > 0.2 Å^−1^) an upward turn of the curve indicated some flexibility in the dimeric structure.

In contrast to the Kratky plot for monomeric A3B_CTD_* having a peak maximum at around *q* ~ 0.10 Å (Figure 3D), the peak maximum of Kratky plot of the A3B_CTD_* dimer was around *q* ~ 0.07 Å, further indicating that the A3B_CTD_* dimer species were hydrodynamically larger in size. This observation was also consistent with the *P*(*r*) plot, which displayed a histogram curve that was slightly skewed (Figure 8E versus Figure 3E) consistent with standard compact dimers. Both the Guinier and *P*(*r*) calculations resulted in very similar *R*g and *I*(0) values (Table 1). The *R*g, *D*_max_, and Porod volumes of the A3B_CTD_* dimer were also indicative of a bigger molecule (1) as compared to the structural parameters derived for the monomeric A3B_CTD_* (Table 1 and Figure 3). Finally, calculation of the MW using the Porod volume generated a MW of ~44 kDa, similar to the MW derived from *I*(0) of ~46 kDa. The derived MW of this protein was roughly twice the size of the monomeric A3B_CTD_*, verifying that this species was consistent with the dimeric A3B_CTD_* form.

Space-filled envelope models were generated from the scattering profile (DAMMIF) of the A3B_CTD_* dimer using P2 symmetry, based on the dimeric state of the protein. These models were averaged resulting in a model with an acceptable NSD score of 0.816 [71,72]. The generated envelope model was first fitted with two A3B_CTD_-QM-E255A monomer crystal structures (PDB: 5CQI [62]). However, the relative orientations of the two A3B_CTD_ molecules were difficult to determine. Therefore, the simulated interface models described earlier in Figure 7 were used. The interface models were compared to the A3B_CTD_* dimer scattering data using rigid body modelling (FoXS [73,74] and CRYSOL [66]) as described in Appendix A. Model 1 resulted in very good visual fit to the data (see Figure 9A); in contrast, model 2 had slightly less optimal visual fits, especially in the mid-*q* region 0.1–0.2 Å^−1^ (Figure 9B). This SAXS fit confirmed that model 1 was the more plausible solution in accordance with the interface prediction analysis discussed above (Δ*G* ~ −38 kcal/mol, Figure 7A). Chi^2^ values for both models did not discriminate, as the Chi^2^ values are dominated by the exponentially more intense low *q* (< 0.1 Å^−1^) data (Figure 9A,B).

Interestingly, the model 1 interface was found to be organized in a similar manner to a previously reported crystal structure of an A3A crystallographic homodimer (PBD 4XXO [52]). In this structure a dimerization groove was formed through the N-terminus slightly away from the active-site, leading to association of the two A3A molecules [52] (10B) in a manner similar to that predicted for the model 1 interface of the dimeric A3B_CTD_*. The A3A dimerization interface was reported to occur through interactions between residues in loop 1, α1, α2, and β2 (Figure 10A). In particular, the interaction of a pair of His56 that were bridged by an exogenous Zn^2+^ ion was noted to be required for dimerization of A3A. Interestingly, Cys239 of A3B_CTD_* is isostructural to His56 of A3A. Superimposing the generated envelope model with the interface dimer model 1 resulted in a good visual fit (Figure 10B). This model provides a good starting point to look at the association of subunits, which may aid in understanding how various conformations may regulate the A3s’ catalytic activity as proposed in earlier studies [25,48,49,50,51,52].

### 3.7. Evaluation of A3B_CTD_* Dimer with dZ-Oligo

The A3B_CTD_* dimeric form was found by the fluorescence assay to be catalytically active, with intermediate deamination activity levels between the monomeric A3B_CTD_* and the weakly active A3B_CTD_-QM-ΔL3 variants (see Appendix A in Appendix A), consistent with a somewhat occluded substrate-binding site. SAXS of this dimer in complex with dZ-oligo (to prevent complications of activity) was examined to understand how the A3B_CTD_* dimer permits catalysis and binding to the ssDNA (Figure 11 and Figure 12). For the A3B_CTD_*/dZ-oligo sample, initial inspection of the UV elution profile (Figure 11C) and the SAXS profile plot (Figure 12A) showed that the *R*g values were not constant across the profiles, signaling that the sample was not homogeneous. Only the peak corresponding to the dZ-oligo had a constant *R*g value of ~10 Å, as expected (711–750 s, Figure 11B,C, frame numbers 668–698, Figure 12). In the presence of the dZ-oligo, the intensity of the peak corresponding to the A3B_CTD_* dimer decreased significantly (elution profile 550–640 sec, Figure 11C, SAXS profile frames 520–610, Figure 12A). Interestingly, a peak coinciding with the relative elution position and size (*R*g ~ 18–19 Å) of the monomeric A3B_CTD_* species appeared upon the decrease of the A3B_CTD_* dimer peak (elution profile shoulder peak 650–700 s (insert Figure 11C), and SAXS profile frames ~ 613–653 in Figure 12A). The three peaks observed in Figure 11B reflected the fact that the A3B_CTD_*/dZ-oligo sample contained three scattering species, which we attribute to A3B_CTD_* dimer species, an A3B_CTD_* monomeric species, and the unbound dZ-oligo (termed species 1–3, respectively). From the UV elution profile, it was observed that the peak maximum associated with the unbound dZ-oligo slightly decreased (~5-fold, Figure 11C) when compared to the control oligonucleotide elution profile (Figure 11B). This decrease suggested that a protein-ssDNA complex may have formed, but it was unclear from the elution profiles which A3B_CTD_* species was forming a complex with the ssDNA (Figure 11C).

As components of each peak overlapped slightly, the boundaries of each scattering species were deconvoluted using SVD/EFA BioXTAS RAW [67]. The singular value decomposition (SVD) function defined that there were three distinct scattering species in the sample (see Appendix A in Appendix A), as previously presumed from the *R*g pattern of the SAXS profiles (Figure 11C and Figure 12A). Using this information, the evolving factor analysis (EFA) method was used to define the boundaries of each species (Figure 12B) and extract their respective 1D-scattering curves (Figure 13). Due to the limitations of SAXS resolution (> 10 Å), the scattering of species 3, which corresponds to unbound dZ-oligo (~9 Å), was not further analyzed. So, focus was placed on the analysis of species 1 and 2. First, the double log plot of log *I*(*q*) vs log *q* indicated that both species were homogeneous as the data at low *q* plateaued (Figure 13B). The Kratky plots reinforced that species 1 was larger than species 2 (Figure 13D). Based on the positions of the peak maxima observed in Figure 13D, species 1 was indicative of a dimeric A3B_CTD_* species (*q* ~ 0.07 Å^−1^) as observed in Figure 8D, while species 2 was suggestive of an A3B_CTD_* monomeric species (q ~ 0.1 Å^−1^) as observed earlier in Figure 3D. The Guinier distribution analysis showed that the scattering profile of species 1 had a reasonably good linear fit, after eliminating the first several lowest *q* data points (Figure 13C), with estimated *R*g and *I*(0) values being in good agreement with values calculated from the *P*(*r*) plot (Figure 13E, Table 1).

The scattering data of species 2 were particularly noisy (Figure 13A), due to its low presence (see Figure 12); therefore, calculation of its structural parameters was less precise than that for species 1. However, the obtained structural parameters of species 2 (*R*g ~ 20 Å, *D*_max_ ~ 60 Å, and MW ~ 23 kDa were comparable with the parameters noted for the monomeric A3B_CTD_* in complex with ssDNA (Table 1). On the other hand, the values of *R*g, *D*_max_, and MW of species 1 were found to be similar to those obtained for the ligand-free A3B_CTD_* dimer (Table 1). This suggested that in the presence of the dZ-oligo the A3B_CTD_* dimer does not form a complex with ssDNA under the SEC conditions that led to a complex of dZ-oligo with monomeric A3B_CTD_*. To examine this notion, the A_260_/A_280_ ratio of the ligand A3B_CTD_* dimer was compared to the ratio of the A3B_CTD_* dimer/dZ-oligo sample to assess the presence of DNA in the eluted protein sample. It was observed that the ratio remained relatively unchanged in the region where species 1 eluted off the SEC column (see Figure 13F), implying that the A3B_CTD_* dimer (species 1) was in fact not bound to the oligo. However, in the region where species 2 eluted off the column, the A_260_/A_280_ ratio of the A3B_CTD_* dimer/dZ-oligo sample (1.335 at 680 s) was significantly larger than the ratio of the ligand-free A3B_CTD_* dimer sample (0.811 at 680 s), consistent with earlier observations that only the monomeric A3B_CTD_* formed a complex with the dZ-oligo. As the A3B_CTD_* dimer species (species 1) does not bind ssDNA in solution under conditions where the monomeric protein does, we conclude that the dimer species was not associated with the observed catalytic activity (see Appendix A in Appendix A). This conundrum is discussed below. Given the similarity of parameters derived from the SAXS data, and the limited resolution, we did not pursue envelope reconstruction or modelling of the deconvoluted data for the A3B_CTD_* plus dZ-oligo mixture.

## 4. Discussion

Here, we report the first SAXS models of a catalytically active A3B_CTD_ in complex with its inhibitor, a dZ-containing ssDNA. Our envelope model indicates that dZ-containing ssDNA is bound to the A3B_CTD_*, while the 3′ end of the oligonucleotide remains rather flexible (Figure 4). This low-resolution model was found to be comparable to the previously reported crystal structure of substrate ssDNA bound to inactivated A3B_CTD_*-E255A mutant (PDB, 5TD5 [30]) (Figure 1 and Figure 4). This demonstrates that our dZ-containing inhibitor binds in the active site of A3B_CTD_ in a similar manner to the dC substrate, supporting the proposed inhibition mechanism [59], where ssDNA forming a U-shape delivers dZ into the active site and in this way, as a competitive inhibitor, blocks the catalytic activity of A3B_CTD_.

Under our conditions, the ligand-free A3B_CTD_* was detected to elute from the SEC column as two species, one of which was the monomeric A3B_CTD_* and the other being of a larger size. Using SEC-SAXS, we were able to elucidate that this second eluting species was in fact an A3B_CTD_* homodimer in solution. A model dimeric structure generated by PRISM2.0 placed two cysteines in sufficiently close proximity to stabilize the otherwise weak interface by a covalent disulfide bond. Interestingly, this interface mimics a previously reported A3A crystallographic dimer (PDB, 4XXO [52]), which is not surprising as our A3B_CTD_* variant contained loop1 from A3A; loop 1 was involved in the formation of this interface. This model fit the SAXS data well, whereas an alternate model, which made much less chemical sense, did not. However, examination of our A3B_CTD_* dimer in the presence of the dZ-oligo revealed that, contrary to the observed weak catalytic activity of the protein, (see Appendix A in Appendix A) the dimer was in fact not bound to the dZ-oligo. 

We always observed a small amount of monomeric A3B_CTD_* present in solution with the dimeric species, even after purification of the dimer species. Conversely, the monomer species always showed some dimer (see Appendix A in Appendix A). The observed catalytic activity is ascribed to this monomer species. After lysis and Ni^2+^-affinity purification, the protein was kept in buffers containing 2 mM β-ME. At biological standard conditions, β-mercapto-ethanol has a standard reduction potential of −220 mV. Cysteine disulfide bridges have potentials in the range −95 to −470 mV [86], depending on environment, with more negative potentials associated with hydrophobic burial. In addition to kinetic aspects associated with burial, the perplexing appearance of dimer in purified monomer and monomer in purified dimer suggests that here the two potentials are closely balanced, such that the change in ionic strength on addition of dZ-oligo tips the balance slightly towards reduction of the protein disulfide link, leading to observable quantities of the monomeric A3B_CTD_*/dZ-oligo species when the dimer is treated with dZ-oligo. Proteolysis and mass-spectrometry of fragments may yield definitive proof of our proposed disulfide link.

The dependence of the protein’s multimeric state on the presence of substrate and other conditions has been noted for A3G in the past [87]. Based on our observations, we propose that the catalytic activity of the enzyme could be regulated upon dimerization, as suggested by earlier studies [25,48,49,50,51,52], controlled possibly by redox stress in the cell, noting that in vivo reduction potential in cell can range from approximately −200 to −350 mV depending on environment and organelle.

## Data Availability

Data are contained within the article or Appendix A.

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
