# Peer review of "Small-Angle X-ray Scattering Models of APOBEC3B Catalytic Domain in a Complex with a Single-Stranded DNA Inhibitor"

_viruses, 2021, doi:10.3390/v13020290_

Round 1

Reviewer 1 Report

In the manuscript titled “Small-angle X-ray scattering models of APOBEC3B catalytic domain in a complex with a single-stranded DNA inhibitor” the authors present SAXS-derived models of mutant/chimera A3B CTD with and without a small catalytically-incompetent ssDNA oligo comprising dZ. The usage of SAXS for solution-based studies of APOBECs is nearly unprecedented and represents a commendable effort to characterize the shape of A3B CTD and to interrogate a potential dimeric interface. The SAXS experiments and the experimental data analysis appears sound and rigorous. However, the only solid conclusions drawn from the data are simply confirmatory of conclusions drawn from other structural studies on A3B or A3A. The manuscript does not make a significant contribution to the APOBEC field. The analysis of the potentially contrived dimeric species of the A3B CTD at the end of the manuscript and the effects on its oligomerization state upon addition of a short ssDNA oligo introduces more confusion and questions without shedding light on the relevance of ssDNA to the oligomeric state of the truncated A3B protein. For these reasons and those noted below, enthusiasm for the manuscript is reduced.

An inherent issue with studying nucleoprotein complexes with SAXS is that NA and protein have different electron densities and thus different inherent scattering factors. It is unclear how the authors addressed this issue during SAXS data collection, analysis and modelling. If the authors opinion is that these differences are negligible due to the small size of the dZ inhibitors than this should be described in detail with supporting references.

The NSD values for both the dZ-A3B and A3B dimer are quite high (above 0.8) and draws into question the validity of the resulting averaged structures. How can the authors justify the usage of generated average models? The assertion that NSD </= 0.8 is acceptable should be supported with specific references.

The relevance of the disulfide linked dimeric A3B-CTD structure is elusive. Is this disulfide linkage physiologically relevant? Or a result of recombinant purification system? A biologically relevant disulfide linked A3 dimer would be unprecedented and would require further validation.   

The final section of the manuscript, presenting SAXS and SEC data of the A3B CTD dimer with and without the dZ oligo is confusing. Addition of the oligo appears to reduce the proportion of A3B dimer species, but the dZ oligo does not appear to be associated with the monomeric species either. There is no resolve to the data presented here.

Line 314: how does the SAXS data specifically support conformational flexibility?

Line 337-340: This is a wholly unsubstantiated statement vaguely referring to a trend observed for all A3s without supporting evidence from the literature. It is unclear what message the authors are attempting to convey.

Why was an A3A-A3B chimera CTD construct used for expression/SAXS analysis instead of WT A3B-CTD? This should be explained fully in the introduction section. What is the basis for using this particular chimera?

Scheme 1: use of the term APOBEC3 here is vague and begs the question which A3’s have actually been tested?

Line 67: The affinity of the dZ oligo for A3B-ctd should be described in greater detail than ‘low uM binding affinity for active A3 enzymes’  

In general, the manuscript would benefit from tighter more concise language.

Reviewer 2 Report

The manuscript describes the first putative structure of active antiviral editing deaminase with an inhibitor dZ-oligonucleotide. The model is based on SAXS solution data of chimera of heavily mutated C-terminal half of APOBEC3B (A3B) with four amino acid changes, deletion of loop 3, and insertion of loop 1 from the related enzyme, APOBEC3A (A3A). The active enzyme model appeared to be similar to the structure of the catalytically dead mutant, so no much novel information was gained. The authors quite extensively addressed the dimerization of the chimera, but it might be attributed to the presence of the loop1 of A3A. The structure dimer of A3A was reported previously, as well as the idea of how dimerization can be involved in the regulation of enzyme activity. The work is technically advanced, but the description is pretty dense (13 Figures plus supplemental materials) and methodology-oriented. The details are of interest to structural biologists. The authors did not discuss how their findings relate to the antiviral properties of A3s. The idea that redox stress of the cell might affect the regulation of A3s dimerization is interesting but needs experimental proof.

Comments.

Short 40-lane Introduction somehow covers a large slice of literature (58 references!), so it is addressed to ultimate A3 scholars.  Maybe, a carton summarizing what the authors need to convey would be nice to see here.

Line 37. What is meant by "A3 members are expressed…"

  1. It would be good to explain better what in under the phrase "…is said to be …but is believed…"
  2. Are all these twelve references are necessary?

103-105. We enjoyed the detailed description of how PAAG gel is prepared.

  1. "…identified to be catalytically active?"
  2. Section "Authors contributions" is blank.

Reviewer 3 Report

Overall a very nice SAXS-based article describing an important biological question.

My main concern is that the entire manuscript is based on a single technique and as we all know, SAXS is a low-resolution complementary technique to other biophysical methods. I would encourage to obtain the molecular weights and monomer-dimer detection using other methods as SEC-MALS, and/or analytical ultracentrifuge. This would strengthen the paper significantly.

Another concern is the dimeric interface. Authors have performed computational calculations to identify high-probability dimers and screened them through solution scattering data. In principle this is fine. However, the next step would be to validate the dimeric interface using side-directed mutagenesis to validate the dimer formation.

Additional comments/suggestions.
1, reference for Mw~Porod volume x 0.6 is required.
2, details about I(0)-based molecular weight determination is required.
3, Almost all SAXS data figures are low-resolution. It seems that the authors used the output figures from ATSAS/BioXTAS. I would recommend exporting raw data and prepare publication-quality figures. Currently, it is hard to follow Guinier plots, raw-data plots, Kratky plots, etc. They did edit the legends to the x- and y-axis, but the figures are not of good quality.
4, for P(r) plots, X-axis is missing units. Y-axis has meaninglessly huge numbers - e.g. 0.000000 and 0.0000005. I would only put 0, 0.5, and so on as these are arbitrary numbers, and write 10^-5 * P(r) as a legend to Y-axis.

Round 2

Reviewer 1 Report

The authors have appropriately and sufficiently addressed the concerns from the reviewers.

Perhaps I've missed it but I cannot find where the authors have defined 'QM' as part of the A3B variant abbreviation in the Intro.

Reviewer 3 Report

I am okay with the revisions, however, some SAXS figures are still low-res and have small fonts. I very much disagree with the response from the authors that SAXS is no less precise than AUC or SEC-MALS for molecular weight determination. I encourage them to read more about AUC and SEC-MALS and find out the fundamental differences with SAXS, in terms of molecular weight determination. There are many excellent reviews/articles out there.